

# The representation of contextual cue is stimulus-specific yet its expression is flexible

Xiaoyu Chen[1], Shuliang Bai[2], Qidan Ren[3], Yi Chen[3], Fangfang Long[3] and Ying Jiang[3]

[1] Research Center of Brain and Cognitive Neuroscience, Liaoning Normal University, Dalian, Liaoning, China
[2] Department of Psychology, Nanjing University, Nanjing, Jiangsu, China
[3] School of Psychology, Guizhou Normal University, Guiyang, Guizhou, China

Corresponding authors
Fangfang Long,
Alongfangfang@163.com
Ying Jiang, jyingpsy@126.com

## ABSTRACT

**Background:** Contextual cueing refers to the phenomenon in which individuals utilize frequently encountered environmental contexts, comprised of distractors, as cues to expedite a target search. Due to the conflict between the widespread occurrence of contextual cue transfer and the observed impact of changing the identity of distractors on contextual cue learning, the content of contextual cue representations remains contentious. Considering the independent nature of contextual cue learning and expression, our proposition is twofold: (1) Contextual cue representations are stimulus-specific, and (2) their expression is highly flexible.
**Methods:** To validate the model, two experiments were conducted. Experiment 1 aimed to confirm the hypothesis that contextual cue representations are stimulus-specific. We manipulated the identity consistency of distractors within repeated scenes during contextual cue learning. Difficulty in contextual cue learning under the identity-changing condition would suggest the necessity of identity within contextual cue representation, indicating the stimulus-specific nature of these representations. Experiment 2 was designed to affirm the conclusion of Experiment 1 and explore the flexibility in the expression of contextual cue representations. This experiment comprised two phases: learning and testing. During the learning phase, participants were exposed to two sets of repeated scenes in different colors under two learning conditions: load and no-load. Working memory load was introduced to interfere with the expression to prevent it from becoming automatic. In the subsequent testing phase, the colors of the two scene sets were interchanged to impede retrieval based on identity. If both load and no-load conditions demonstrate similar levels of contextual cue effects during the testing phase, it implies the flexibility in the expression of contextual cue representations and confirms the conclusion of Experiment 1.
**Results:** In Experiment 1, a notable contextual cue learning effect was observed under the identity-consistent condition ($p = 0.001$). However, this effect was not evident under the identity-changing condition ($p = 0.286$). This finding strongly supports the stimulus-specific nature of contextual cue representation. In Experiment 2, the contextual cueing effect appeared but did not show a significant difference between the two conditions ($t(23) = 0.02$, $p = 0.987$, $BF_{10} = 0.215$), indicating the cognitive system's ability to flexibly redefine retrieval cues. This adaptability aligns with our hypothesis and confirms the high flexibility in the

expression process of contextual cue representations and confirms the conclusion of Experiment 1.

# INTRODUCTION

Contextual cueing, first observed in real-world settings and later replicated in laboratory experiments by *Chun & Jiang (1998)* through visual search tasks, involves utilizing repeatedly occurring environmental contexts as cues to aid target search (*Jiang & Sisk, 2020*). In their experiments, participants encountered two types of search scenes: novel and repeated. Novel scenes were generated randomly before each trial, whereas repeated scenes remained constant throughout the experiment. The discovery was made that repeated scenes resulted in shorter reaction times compared to novel scenes, termed the contextual cueing effect. This effect was attributed to subjects learning and utilizing the repeated distractor context as a cue to enhance target search efficiency. Their experimental approach established the foundational paradigm of visual statistical learning known as the contextual cueing paradigm (*Goujon, Didierjean & Thorpe, 2015*). Notably, participants were unable to consciously recognize these repeated scenes, indicating that contextual cueing within this paradigm is an implicit learning phenomenon.

Considerable controversy surrounds the content of contextual cue representation, primarily concerning the role of distractor identity alongside spatial position within search scenes. Central to this debate is whether the identity of distractors constitutes an essential element of contextual cue representation. Three distinct perspectives, supported by evidence, have emerged on this matter.

In their seminal study, *Chun & Jiang (1998)* first instructed participants to learn a set of contextual cues. Subsequently, without altering the spatial layout of the learned repeated scenes, they manipulated the identity of items (Experiment 2). The results revealed that participants still demonstrated the contextual cue effect. Consequently, they concluded that non-spatial information within the scenes does not constitute a component of the contextual cue representation.

*Makovski (2016)* objected to this notion. They investigated the content of contextual cue representation using real-world objects as stimuli. Their research revealed that subjects only demonstrated contextual cue learning when both the position and identity of items were repeated. Subsequent studies replicated these findings (*Makovski, 2017*), providing evidence that item identity information is indeed a crucial component of contextual cue representation.

*Jiang & Song (2005)* proposed a compromise perspective, suggesting that the inclusion of item identity in contextual cue representation depends on its facilitation of retrieval. In their experiment, subjects underwent two consecutive phases: learning contextual cues and testing learning outcomes. Similar to the findings of *Chun & Jiang (1998)*, they observed a contextual cue effect even when scenes changed color or shape between learning and

testing phases, indicating transfer (Experiment 3 for color, Experiment 1 for shape). However, when subjects learned two sets of scenes with different identity features but were tested with unified identity features, there was no transfer of contextual cues (Experiment 2 and Experiment 4). This led to the conclusion that the cognitive system selectively incorporates identity information to narrow retrieval scope.

Both *Chun & Jiang (1998)* and *Jiang & Song (2005)* observed transfer phenomena in contextual cueing. Transfer phenomena have been widely documented in visual statistical learning (*Turk-Browne & Scholl, 2009*), with contextual cueing even exhibiting cross-modal transfer capabilities (*Nabeta, Ono & Kawahara, 2003*). The transfer commonly associated with successful representation retrieval in new contexts (*Royer, 1979*). However, few studies have provided a retrieval perspective to explain contextual cueing transfer. Additionally, the learning and expression (including retrieval) of contextual cues are independent processes (*Annac et al., 2013*; *Jiang & Chun, 2001*; *Jiang & Leung, 2005*; *Manginelli et al., 2013*; *Travis, Mattingley & Dux, 2013*), suggesting that transfer does not solely depend on representation content. Moreover, in light of findings by Makovski supporting the importance of item identity, transfer in *Jiang & Song (2005)* and *Chun & Jiang (1998)* may reflect the cognitive system's selective retrieval cue utilization rather than representation content.

We propose a novel explanation to reconcile the contradictory findings. This explanation consists of two primary points. Firstly, contextual cue representations are stimulus-specific, encompassing spatial and identity information of distractor contexts within repeated scenes. Secondly, the expression of contextual cue representations is highly flexible, selectively utilizing scene information as retrieval cues to enhance retrieval efficiency. To validate our hypotheses, we conducted two experiments.

Experiment 1 aimed to validate our first hypothesis. Previous research by *Makovski (2016)* utilized real-world object images as stimuli and demonstrated the significance of item identity in contextual cue representation. However, these objects inherently trigger prioritized and automatic semantic processing over other simple features (*Flaudias & Llorca, 2014*; *Luo, 1999*), potentially confounding the active learning of distractor identity within contextual cue representation. Methodologically, we adopted a paradigm akin to *Makovski (2016)*'s study but employed materials with ambiguous semantic content to mitigate automatic processing interference. Failure to observe contextual cue learning under these conditions would suggest the stimulus-specific nature of contextual cue representation.

If Experiment 1 confirms stimulus-specific representations, the transfer observed by *Chun & Jiang (1998)* and *Jiang & Song (2005)* could support the flexibility in retrieval cue selection. However, *Jiang & Song (2005)* also noted that when two sets of repeated scenes with different identity features were learned, changing the identity feature in the testing phase led to the disappearance of the contextual cue effect. This result has two potential explanations: firstly, the cognitive system may have formed an automated representation expression pattern during cue learning, which could not be promptly abandoned in the testing phase; secondly, contextual cue representations may only need to include information limited to individual representations, where color may not be essential.

Clearly, the second explanation would constrain the conclusion of Experiment 1, given their small sample size which could introduce considerable statistical variability, warranting a reevaluation of their findings.

Experiment 2 aims to verify their findings and explore the flexibility in contextual cue expression. Similar to *Jiang & Song (2005)*, Experiment 2 consisted of two unprompted task phases, where participants encounter two sets of repeated scenes in different colors during the learning phase. During the learning phase, participants would learn two sets of repeated scenes in different colors. In the testing phase, unlike *Jiang & Song (2005)*'s approach, the colors of the two sets of learned scenes would be swapped to prevent any potential confusion caused by unifying them into one color. Additionally, Experiment 2 introduced a working memory load to selectively interfere with expression during cue learning (*Annac et al., 2013*; *Manginelli et al., 2013*; *Travis, Mattingley & Dux, 2013*), preventing the formation of automated expression processing. With an increased number of participants, if no transfer occurs under the no-load condition but does occur under the load condition, it suggests that contextual cues are stimulus-specific, but their expression becomes automated during learning, hindering their expression after feature changes. If transfer is observed under both conditions, it upholds the conclusion of Experiment 1 and further demonstrates that contextual cue expression does not become automated, indicating flexibility.

**Experiment 1: contextual cue representations are stimulus-specific**

In the classic contextual cue paradigm proposed by *Chun & Jiang (1998)*, participants engaged in a visual search task featuring novel and repeated scenes. These scenes comprised abstract shapes resembling "T" and "L" letters formed by lines. The randomly rotated "L" acted as the distractor, while the "T" served as the target. Experiment 1 replicated this setup to prevent automatic semantic processing, defining identity operationally as the rotation angle of each "L" distractor. Two experimental conditions for repeated scene identity were established: change and consistent. In the change condition, distractors' spatial layout remained consistent while their identity varied across repetitions. Conversely, the consistent condition ensured consistent spatial positions and identity for distractors within repeated scenes. If subjects' contextual cue learning is impeded under the change condition, it indicates the essential role of distractor identity in contextual cue representation, affirming stimulus-specificity.

## MATERIALS AND METHODS

### Participants

This experiment involved 22 healthy, right-handed adult participants (14 females; mean age = 21.86, *SD* = 1.67) with normal or corrected-to-normal vision. Participants had no history of mental illness and had not engaged in any contextual cue paradigm experiments within the past week to prevent potential proactive interference in contextual cue learning (*Mednick et al., 2009*). All participants provided informed consent and received compensation upon completion of the study. Ethical approval was obtained from the

School of Psychology, Guizhou Normal University (Approval number: GZNUPSY. N.202309E [0019]).

## Apparatus

The experiment was conducted using Matlab (The MathWorks, Inc., Natick, MA, USA) with the PsychoToolbox extensions (*Brainard, 1997*), displayed on a Dell 19-inch LED monitor with a refresh rate of 60 Hz and a resolution of 1,024 × 768. Participants viewed the display from approximately 60 cm without restraint.

## Stimuli

The task employed stimuli resembling "L" and "T" shapes, composed of black lines (RGB = 0, 0, 0), against a gray background (RGB = 120, 120, 120). Each shape subtended a visual angle of 0.8° × 0.8° and appeared within an invisible rectangle (16.0° × 12.0°) at the screen's center, divided into an 8 × 6 grid. Stimuli were distributed evenly across the grid cells, with one target ("T" rotated randomly 90° left or right) and 12 distractors ("L" rotated randomly 0°, 90°, 180°, or 270°) in each search scene. Distractors were evenly distributed across four quadrants centered around the scene's midpoint to prevent clustering effects. To prevent collinearity, each stimulus was randomly displaced by 0.1° from the grid center in any direction.

Before the task, the program randomly determined 16 target positions, with eight designated for repeated scenes and eight for novel scenes. These positions were evenly distributed across quadrants to prevent spatial attention biases. Eight repeated scenes were randomly generated, maintaining fixed distractor positions throughout. Each participant encountered a unique target set and repeated scenes. In the identity-changing condition, distractor positions remained constant while their identity, defined by rotation angle, varied randomly before each trial. Conversely, in the identity-consistent condition, both distractor positions and identities remained consistent throughout the experiment (Fig. 1A). For novel scenes, both positions and identities of distractors were randomized before each trial, varying unpredictably throughout the experiment.

## Trial procedure

This study utilized the standard contextual cue paradigm (*Chun & Jiang, 1998*; *Olson & Chun, 2001*), requiring participants to engage in a visual search task. Each trial began with a 1-s blank screen containing a central fixation cross (1.0° × 1.0°). Subsequently, a search display appeared, wherein participants searched for the target "T" among "L" distractors and identified the "T"'s orientation. Participants indicated if the "T" was rotated 90° left by pressing the "F" key or 90° right by pressing the "J" key. Participants were instructed to respond quickly and accurately. The search display remained visible until a response was made.

## Experiment procedure

Given the significant impact of repeated *versus* novel scene ratios on contextual cueing (*Zinchenko et al., 2018*), maintaining an equal number of repeated and novel scenes in both learning conditions was essential. To achieve this balance, participants completed two

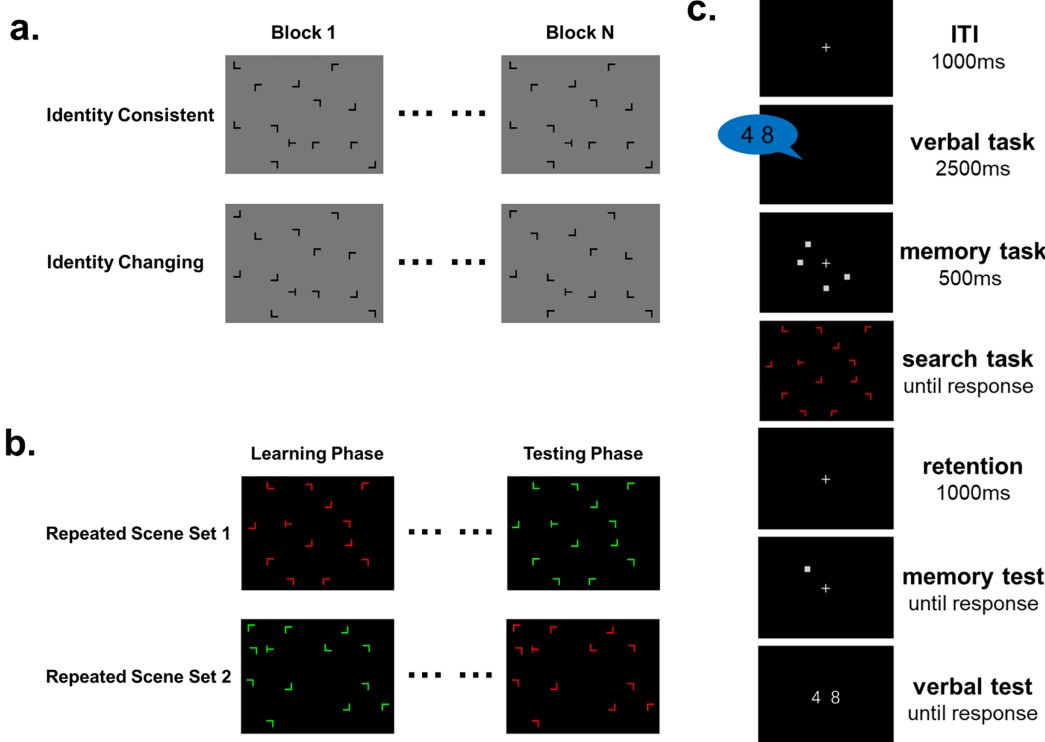

**Figure 1 Experiment stimuli and procedure schematic.** Each search scene consisted of 12 distractors and one target. The target was a rotated "T", and the distractors were rotated "Ls", all constructed from the same line segments. (A) In Experiment 1, Participants completed two identity conditions: identity consistent and identity changing. Identity is defined as the rotational angle of the distractors that form the scene. Identity consistent condition involved maintaining consistent spatial positions and identities of the distractors across the experiment. Identity changing condition involved consistent spatial positions of the distractors while the identities changed across the experiment. (B) In Experiment 2, the experiment was divided into learning and testing phases. Participants engaged in visual search tasks in the learning phase with two sets of repeated scenes containing items of different colors to learn contextual cues. The testing phase involved swapping the colors of the repeated scenes. (C) Experiment 2 introduced interference conditions using a dual-task interfering with the expression of contextual cue representations during the learning phase. Participants performed a visual search task during the working memory retention period.

experiments, separately engaging in identity-consistent and changing conditions. To counteract sequence effects, half of the participants began with the changing condition, while the other half started with the consistent condition. To minimize proactive interference in contextual cue learning, participants observed a minimum three-day interval between the two experimental conditions (*Mednick et al., 2009*).

Each experiment comprised a practice task and the formal experiment. In the practice task, participants completed 16 search trials with randomly generated visual scenes to familiarize themselves with the task. Subsequently, the formal experiment commenced. Each task block included eight repeated scenes and eight novel scenes presented randomly. Participants completed 30 blocks of visual searches in each identity condition, totaling 480 trials. Breaks were permitted every 80 trials, with participants able to end breaks by pressing the space bar.

## Statistical analysis

Initially, we computed accuracy rates for all participants in both learning conditions. Regarding reaction times (RTs), incorrect trials were first excluded, followed by the removal of trials deviating beyond 2.5 $\sigma$ from the mean and those with RTs under 200 ms. The remaining trials were considered valid. For the changing condition, the average number of valid trials retained was 460.59 ($SD = 6.28$), while for the consistent condition, it was 458.04 ($SD = 12.21$). Given the small sample sizes of eight repeated and eight novel scenes per block, data were aggregated from five blocks into an epoch to mitigate statistical fluctuations, following standard practice in contextual cueing paradigm studies (*Chun & Jiang, 1998*; *Olson & Chun, 2001*).

We conducted a three-way within-subjects repeated-measures ANOVA using mean RTs as the dependent variable, with factors of Identity (changing *vs.* consistent), Scene type (repeated *vs.* novel), and Time course (epochs 1~6). An interaction between Identity and Scene type, where the contextual cueing effect in the consistent condition surpasses that in the changing condition, would suggest the importance of distractor identity information in forming the contextual cue representation.

# RESULTS

## Accuracy

In the changing condition, participants achieved an average accuracy of 98.60% ($SD = 1.13\%$), compared to 98.21% ($SD = 2.37\%$) in the consistent condition.

## Reaction times

The RT results are presented in Fig. 2A. A three-way within-subject repeated-measures ANOVA was conducted with factors of Identity (changing *vs.* consistent), Scene type (repeated *vs.* novel), and Time course (epochs 1~6). The results indicated a non-significant three-way interaction ($F(3.21,67.54) = 0.92$, $p = 0.439$, Greenhouse-Geisser corrected), as well as non-significant interactions between Scene type and Time course ($F(3.30,69.20) = 1.10$, $p = 0.358$, Greenhouse-Geisser corrected) and between Identity and Time course ($F(3.29,69.15) = 1.62$, $p = 0.189$, Greenhouse-Geisser corrected). However, a significant interaction was observed between Identity and Scene type ($F(1,21) = 5.02$, $p = 0.036$, $\eta^2 = 0.19$). Simple effects tests for the Identity × Scene type interaction revealed a significant RT difference between novel and repeated scenes under the identity-consistent condition ($p = 0.001$, Bonferroni corrected), indicating a contextual cueing effect. Conversely, under the identity-changing condition, no significant RT difference was observed ($p = 0.286$, Bonferroni corrected), as depicted in Fig. 2B.

Furthermore, the main effect of Identity was non-significant ($F(1,21) = 0.05$, $p = 0.830$). However, significant main effects were found for Scene type ($F(1,21) = 15.50$, $p = 0.001$, $\eta^2 = 0.43$) and Time course ($F(3.08,64.86) = 14.85$, $p < 0.001$, $\eta^2 = 0.41$, Greenhouse-Geisser corrected). Analysis of Time course conditions revealed a general shortening of RTs across all epochs, indicating an overall practice effect.
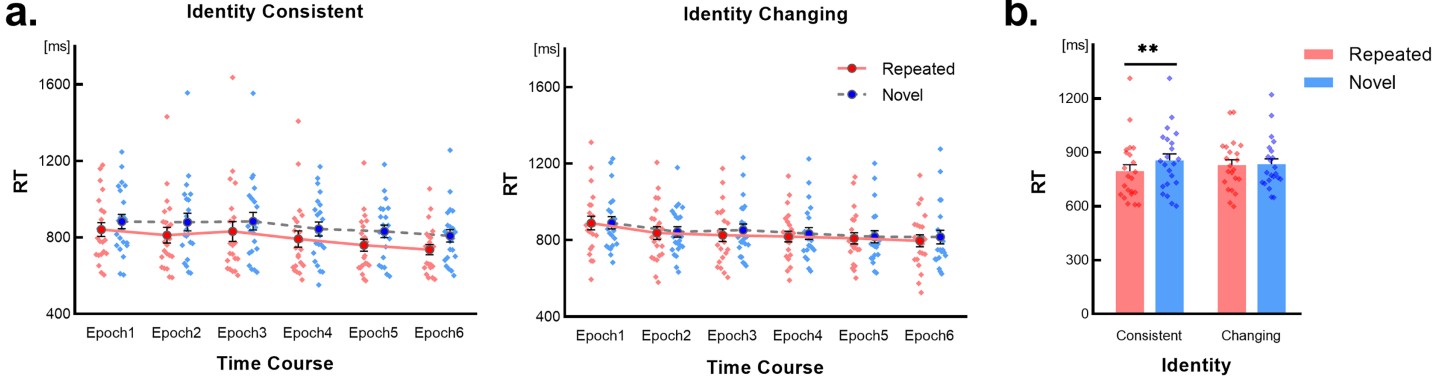

**Figure 2** **Results of Experiment 1.** Error bars represent the standard error of the mean (SEM). Light-colored dots represent individual results. (A) Changes in RTs over time for the identity consistent and identity changing conditions, concerning repeated scenes and novel scenes. (B) RTs for repeated scenes and novel scenes under different identity conditions. "**" Indicates significant simple effect detected with $p < 0.01$, Bonferroni corrected.

We hypothesized that the emergence of the contextual cueing effect is due to learning. However, there was no interaction related to Time course. To address inherent differences between repeated and novel scenes under the identity consistent condition, RTs in the first epoch were subjected to a 2 (Scene type: repeated *vs.* novel) × 5 (Time course: block 1~5) repeated-measures ANOVA. However, we did not find a significant interaction between the two factors, ($F(2.30,48.38) = 0.21$, $p = 0.935$, Greenhouse-Geisser corrected). Nevertheless, we observed significant or marginally significant main effects of Scene type and Time course ($p < 0.07$), indicating inherent differences between the two scene types. As all target locations and repeated scenes' distractors were randomly generated at the beginning of the experimental program, and each participant faced a unique set of target locations and repeated scenes, these results can be attributed to chance.

To ensure contextual cue learning occurred under the identity consistent condition, we conducted a 2 (Scene type: repeated *vs.* novel) × 2 (Time course: epoch 1 *vs.* epoch 6) ANOVA to detect the extension of the time difference between the two scene types. A significant interaction was found ($F(1,21) = 5.44$, $p = 0.030$, $\eta^2 = 0.206$), as well as significant main effects of Scene type and Time course ($p < 0.01$). Pairwise comparison confirmed that the time difference in epoch 1 was shorter than in epoch 6 ($t(21) = 2.33$, $p = 0.030$), indicating contextual cue learning occurred under the identity consistent condition in Experiment 1.

Additionally, we conducted a 2 (Scene type: repeated *vs.* novel) × 2 (Time Course: epoch 1 *vs.* epoch 6) ANOVA on the RTs of the identity changing condition. We found that the main effect of Time course was significant ($F(1,21) = 22.50$, $p < 0.001$, $\eta^2 = 0.517$), but other effects were not significant ($p > 0.1$), indicating a failure in contextual cue learning under this condition.

## DISCUSSION

The results of Experiment 1 showed that under the identity changing condition, participants failed to learn contextual cues despite consistent spatial positions of

distractors within repeated scenes. This suggests that contextual cue learning necessitates both consistent identity and spatial positions within specific repeated scenes. This finding strongly supports the idea that the identity of distractors is crucial for contextual cue representation, suggesting that such representation is stimulus-specific.

## Experiment 2: flexibility in the expression of contextual cue representations

As previously noted, *Jiang & Song (2005)* observed that changing the identity feature in the testing phase led to the disappearance of contextual cue effects after learning two sets of repeated scenes with different identity features. This finding suggests two potential explanations: firstly, the cognitive system may have formed some form of automated representation expression during contextual cue learning and was unable to promptly abandon this expression processing during subsequent testing phases; secondly, contextual cue representations may need to include information limited to individual representations, and color may not be a necessary component of this information. Clearly, the second explanation poses a risk to the conclusions drawn from Experiment 1.

To verify the flexibility of contextual cue representation in expression and to validate the conclusion of Experiment 1, we conducted Experiment 2. Following the approach of *Chun & Jiang (1998)* and *Jiang & Song (2005)*, Experiment 2 comprised two undisclosed task phases: one for learning and the other for testing. Additionally, Experiment 2 featured a larger sample size than *Jiang & Song (2005)*. During the learning phase, subjects were exposed to two sets of repeated scenes with different colors, under load and no-load conditions. In contrast, the load condition required participants to perform visual searches during the retention period of a working memory task to hinder expression, ensuring that expression was not automatic during learning. In the testing phase, the colors of the two learned repeated scenes were swapped. If, as observed by *Jiang & Song (2005)*, no transfer effects are evident under the no-load condition but emerge under the new load condition, it indicates that representations are stimulus-specific, and the expression process is constrained by past experiences. Conversely, if equivalent transfer effects are observed in both load and no-load conditions, it confirms that previous experience does not influence subsequent adjustments of retrieval cues, thus confirming the flexibility of expression and supporting the conclusions of Experiment 1.

## MATERIALS AND METHODS

### Participants

Experiment 2 enrolled 24 healthy right-handed adult participants, averaging 18.75 years old ($SD = 1.07$), with 12 females. The participant selection criteria mirrored those of Experiment 1. All participants provided informed consent and was compensated upon completion of the study. Ethical approval was obtained from the School of Psychology, Guizhou Normal University (Approval number: GZNUPSY.N.202309E [0019]).

### Apparatus

The same equipment as Experiment 1 was utilized.

## Stimuli

The visual search task utilized red (RGB = 255, 0, 0) or green (RGB = 0, 255, 0) shapes resembling letters "L" and "T", each composed of two lines of equal length and thickness. These shapes subtended a visual angle of $0.8° \times 0.8°$ against a black background (RGB = 0, 0, 0). Similar to Experiment 1, stimuli were presented within an invisible rectangle ($16.0° \times 12.0°$) at the center of the screen, divided into an $8 \times 6$ grid, with stimuli appearing at various positions within this grid. Each search scene contained one randomly rotated "T" target (90° left or right) and 12 distractors ("L" rotated randomly at 0°, 90°, 180°, 270°) (Fig. 1B).

Prior to task commencement, the program randomly designated 32 target locations—16 for repeated scenes and 16 for novel scenes—equally distributed across each quadrant without overlap. Sixteen repeated scenes were generated, comprising eight red and eight green items. The spatial layout and identity information, excluding color, remained consistent across these repeated scenes.

## Trial procedure

Experiment 2 comprised a learning phase and a testing phase. The learning phase presented only repeated scenes for participants to acquire contextual cues. Two learning conditions were employed: the load and no-load conditions. The trial flow in the no-load condition mirrored Experiment 1.

In the load condition, a dual-task paradigm involving working memory and visual search was implemented during the learning phase (Fig. 1C). For the working memory task, participants memorized spatial locations. Each trial began with a 1-s blank screen featuring a central fixation point. To prevent verbal encoding, two digits were audibly presented for 2.5 s, chosen randomly from 1 to 9. A 500 ms working memory array then appeared, consisting of four gray squares (RGB = 120, 120, 120), each measuring $0.8 \times 0.8$ degrees. Participants memorized the spatial locations of these items, randomly positioned among eight equidistant locations on an imaginary circle. Following the memory array, the visual search scene appeared. After responding, a 1-s retention period with central fixation followed. A spatial working memory probe array then appeared, requiring participants to match the positions with the initial memory array. Subsequent tasks involved responding to presented digits based on the initial auditory presentation. Participants pressed "F" for a match and "J" for a non-match in both cases.

## Experiment procedure

Similar to Experiment 1, Experiment 2 mandates a three-day interval between the load and no-load conditions for each participant. Participants are evenly split, with half starting with the load condition and the other half with the no-load condition to counterbalance order effects.

Before the formal experiment, participants undergo a 16-trial practice session, mirroring the learning phase, ensuring their readiness.

The formal experiment consists of two uninterrupted phases: learning and testing. In the learning phase, one block includes two differently colored repeated scenes, totaling 16 scenes presented randomly. Experiment 2's learning phase features the load and no-load conditions. In the no-load condition, participants perform regular visual searches, while in the load condition, they engage in a dual-task paradigm of working memory and visual search.

Following the learning phase, the testing phase commences immediately without notice. Regardless of the condition, this phase involves a single visual search task. Each block comprises 16 previously learned repeated scenes and 16 new novel scenes, randomly presented. The colors of the learned repeated scenes are swapped, with each block including eight red and eight green novel scenes.

Each learning condition involves 480 visual search trials. The learning phase consists of 20 blocks (320 trials), and the testing phase comprises five blocks (160 trials).

## Statistical analysis

### Working memory task

We assessed accuracy in both the spatial and auditory working memory tasks under the non-interference condition. A one-sample *t-test* verified performance against the chance level of 50% to ensure task engagement.

### Visual search task

Initially, we computed accuracy in the visual search task for both learning conditions. Regarding reaction times (RTs), akin to Experiment 1, we excluded trials with incorrect responses and those deviating beyond 2.5 $\sigma$ from the mean or with RTs below 200 ms. This left us with an average of 469.58 ($SD = 10.25$) valid trials for the load condition and 473.62 ($SD = 5.00$) for the no-load condition. A paired-sample *t-test* would confirm any significant differences in valid trials between the two conditions ($t(23) = 2.479, p = 0.021$), likely due to varying task difficulties.

Similar to Experiment 1, we treated five blocks as an epoch, utilizing epochs as the smallest unit of analysis in the learning time course. Initially, a within-subject repeated-measures ANOVA analyzed RT data from the learning phase, employing factors of 2 (Learning condition: load *vs*. no-load) × 4 (Time course: epochs 1~4). This analysis aims to determine if the dual-task working memory induces decreased visual search efficiency. Slower RTs in the visual search of repeated scenes under the load condition would suggest a decline in visual search efficiency due to factors such as task switching costs or difficulty in accessing contextual cue representations.

A two-way within-subject repeated-measures ANOVA would be conducted on average RTs in the testing phase, with factors of 2 (Learning condition: load *vs*. no-load) × 2 (Scene type: repeated *vs*. novel). If the cognitive system can flexibly adjust the expression of the representation by selectively choosing retrieval cues, then an equal contextual cueing effect in different learning conditions would be observed.

## RESULTS

### Working memory task

Under the load condition, spatial working memory accuracy averaged 77.45% ($SD$ = 10.18%). A one-sample $t$-test revealed that participants' accuracy significantly exceeded chance level (50%) ($t(23)$ = 13.206, $p < 0.001$). In the digit memory task, average accuracy was 96.32% ($SD$ = 3.64%), with a one-sample $t$-test showing significant performance above chance (50%) ($t(23)$ = 62.280, $p < 0.001$). All participants surpassed 50% accuracy in both memory tasks.

### Visual search task

#### Accuracy

In the visual search task, accuracy averaged 98.67% ($SD$ = 1.04%) under no-load conditions and 97.83% ($SD$ = 2.14%) under load conditions.

#### Reaction times in learning phase

The RTs for each epoch during the learning phase are depicted in Fig. 3A on a white background. We conducted a 2 (learning condition: load $vs$. no-load) × 4 (Time course: epoch 1~4) repeated-measures ANOVA on the RT data during the learning phase. The results revealed a significant main effect for learning condition ($F(1,23)$ = 17.40, $p < 0.001$, $\eta^2$ = 0.43), a significant main effect for time course ($F(2.24,51.54)$ = 75.89, $p < 0.001$, $\eta^2$ = 0.77, Greenhouse-Geisser corrected), and a significant interaction effect ($F$ (2.13,49.01) = 5.08, $p$ = 0.009, $\eta^2$ = 0.18, Greenhouse-Geisser corrected).

*Post-hoc* tests indicated significant RT differences between the two learning conditions for each epoch ($p < 0.05$). This interaction effect stemmed from a non-significant RT difference between epoch 3 and epoch 4 under the no-load condition ($p$ = 1.000, Bonferroni corrected), and a marginally significant difference under the load condition ($p$ = 0.070, Bonferroni corrected). This suggests that introducing a dual-task during the learning phase significantly increased the visual search difficulty and affected the time required to achieve maximal practice effects.

#### Reaction times in testing phase

The results of the 2 (learning condition: load $vs$. no-load) × 2 (Scene type: repeated $vs$. novel) repeated-measure ANOVA on RTs during the testing phase revealed a non-significant interaction ($F(1,23) < 0.001$, $p$ = 0.987). However, significant main effects were observed for the learning condition ($F(1,23)$ = 12.99, $p$ = 0.001, $\eta^2$ = 0.36) and Scene type ($F(1,23)$ = 29.88, $p < 0.001$, $\eta^2$ = 0.56) revealing that regardless of whether the expression of contextual cue representations was disturbed during the learning phase, RTs for repeated scenes during the testing phase were consistently lower than those for novel scenes, indicating the presence of a contextual cueing effect (Fig. 3A in gray background). Additionally, across both scene conditions, the RTs under the load condition were longer than those under the no-load condition, suggesting that the RT extension effect caused by the dual-task during the learning phase persisted into the testing phase.
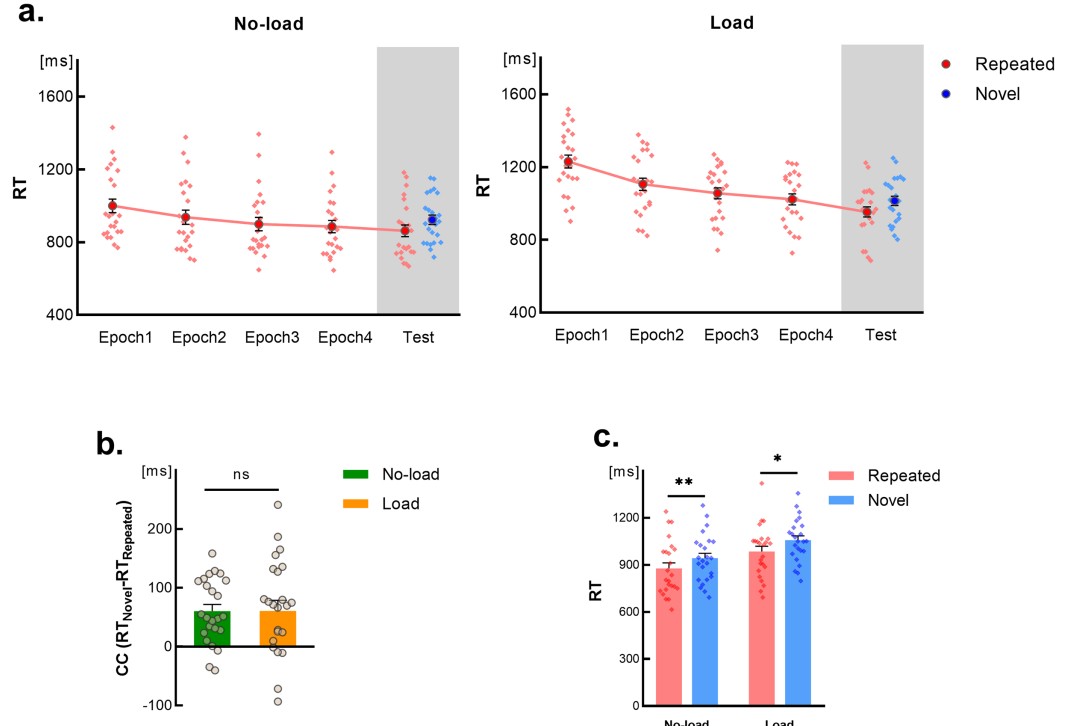

**Figure 3 Results of Experiment 2.** Error bars indicate the standard error of the mean (SEM). Light-colored dots represent individual results. (A) Changes in RTs over time for repeated scenes and novel scenes under two learning conditions. Results from the learning phase are depicted in a white background, while results from the testing phase are shown in a gray background. (B) Contextual cue effects (CC) under two learning conditions. The "ns" above the bars indicates paired-sample *t-test* results with $p > 0.05$. (C) Comparison of RTs for repeated scenes and novel scenes in the first block of the testing phase under two learning conditions. Light-colored dots represent individual subject results. The "*" above the bars denote *post-hoc* test results between repeated scenes and novel scenes for each learning condition, where "*" indicates $p < 0.05$, "**" indicates $p < 0.01$.

The main effect observed in the learning condition suggests the establishment of enduring automatic processing during learning. To ensure that this automatic processing does not relate to the expression of contextual cue representation, the contextual cueing effect (CC) was assessed under both load and no-load conditions. This effect, defined as the difference in reaction times (RTs) between novel and repeated scenes, was analyzed using a paired-sample *t-test*. The results revealed no significant difference in the contextual cueing effect between the two conditions ($t(23) = 0.02$, $p = 0.987$) (Fig. 3B), indicating an equivalent level of contextual cue effect under both learning conditions. A Bayesian paired sample t-test further supported this equivalence according to the classification criteria proposed by *Wagenmakers et al. (2018)* ($BF_{10} = 0.215$). These findings demonstrate that prior experiences do not impact subsequent cognitive systems' adjustments in retrieving cues.

To ensure that the results observed in the testing phase of Experiment 2 are due to the occurrence of contextual cue learning during the learning phase, we conducted a 2 (Learning condition: load *vs*. no-load) × 2 (Scene type: repeated *vs*. novel) repeated-measure ANOVA to test whether there is a contextual cueing effect in the first block. The data analysis revealed a non-significant interaction ($F(1,23) = 0.003$, $p = 0.956$);

however, both the learning condition and scene type exhibited significant main effects ($p < 0.01$). *Post-hoc* tests demonstrated the presence of contextual cueing effects under both learning conditions ($p \leq 0.01$, Bonferroni corrected) (Fig. 3C), confirming that the contextual cueing effects observed during the testing phase were reflective of the learned contextual cue representations.

## DISCUSSION

Experiment 2 demonstrated contextual cue effects in both load and no-load conditions during the testing phase, indicating the cognitive system's flexibility in redefining retrieval cues regardless of opportunities to establish an automatic expression in the learning phase. Moreover, the no-load condition results contradict *Jiang & Song (2005)*, confirming Experiment 1's effectiveness.

Furthermore, we observed that the extended RTs induced by dual-tasking in the learning phase persisted into the testing phase, indicating enduring automatic processing established during learning. However, this automatic processing does not pertain to the expression of contextual cue representation, as no contextual cue effect difference was found between learning conditions. Additionally, Experiment 2 revealed that task difficulty influenced the time to reach the maximum practice effect, especially in the dual-task learning phase, where the practice effect did not peak between epochs 3 and 4.

### General discussion

The study aimed to explore contextual cue representation characteristics related to learning and expression. We hypothesized that these representations are stimulus-specific, incorporating spatial positions and non-spatial identity information within repeated scenes, while their expression is highly flexible, enabling the cognitive system to timely select retrieval cues to enhance retrieval efficiency. To test this, we conducted two experiments. Experiment 1 revealed that when spatial positions remained consistent but non-spatial identity information changed randomly during contextual cue acquisition, it significantly hindered learning. This underscored the importance of distractor identity information in contextual cue representations, highlighting their stimulus specificity. Experiment 2 further demonstrated that despite working memory load during cue acquisition, an equivalent contextual cue effect emerged after scene color swapping. This suggests the cognitive system's flexibility in selecting retrieval cues amid environmental changes. This supports the idea of flexible expression in contextual cue representations.

### Contextual cue representations are stimulus-specific

*Makovski (2016)* acknowledged the importance of non-spatial identity in contextual cue learning but used real-world objects with conceptualizable features, inducing significant automatic semantic processing. Given that conceptual information aids memory more effectively than perceptual distinctiveness (*Konkle et al., 2010*), semantic processing may dominate during contextual cue learning. Our experiments mitigated such interferences and directly confirmed the essential inclusion of non-spatial stimulus identity information within contextual cue representations.

Contextual cues exemplify a classic case of visual statistical learning within implicit memory (*Perruchet & Pacton, 2006*). Participants lack intentional learning, awareness of the process, and cannot overtly recognize scenes enhancing search efficiency (*Cleeremans, Allakhverdov & Kuvaldina, 2019*; *Goujon, Didierjean & Thorpe, 2015*; *Sisk, Remington & Jiang, 2019*). Visual statistical learning, like other forms of implicit cognition, lacks cognitive control and learning strategies, representing a highly automated form of unsupervised learning (*Cleeremans, Allakhverdov & Kuvaldina, 2019*; *Goujon, Didierjean & Thorpe, 2015*; *Turk-Browne & Scholl, 2009*). Moreover, it fosters stimulus-specific representations rather than abstract rule representations (*Conway & Christiansen, 2006*). These traits contribute to the challenge of selectively memorizing specific information.

Previous models of contextual cues have often emphasized the significance of spatial position (*Brady & Chun, 2007*; *Jiang & Wagner, 2004*; *Olson & Chun, 2002*). Spatial position largely dictates the overall layout of distractors within the visual field, while the identity of these distractors usually contains more detailed information. Perceptually, global information takes precedence over details (*Navon, 1977*), and in visual statistical learning, forecasting the overall context outweighs individual components (*Yan et al., 2023*). These findings suggest that spatial position holds more learning value compared to identity. Thus, when distractor identity varies, the cognitive system appears to prioritize learning spatial position to reduce the learning load. However, our empirical evidence does not support this hypothesis. Implicit memory, especially visual statistical learning, is a form of long-term memory (*Cleeremans, Allakhverdov & Kuvaldina, 2019*; *Kim et al., 2009*), which offers virtually unlimited storage potential. Consequently, during implicit memory processes, there is no need to reduce memory load. From this perspective, acquiring all scene information is reasonable.

In summary, while selectively learning specific information may reduce the learning load, empirical evidence indicates that the cognitive system learns all distractor information during contextual cue learning.

## The expression of contextual cue representation is flexible

Experiment 2 demonstrated the seamless transfer of contextual cues between learning and testing phases despite color exchange, suggesting that color was not utilized as a retrieval cue in the testing phase, contrary to *Jiang & Song (2005)*. Additionally, interference in the expression of contextual cue representation did not impact transfer degree, confirming the stimulus-specific nature of contextual cue representations and the flexibility in their expression.

Previous research suggests that the learning and expression of contextual cues operate independently (*Jiang & Chun, 2001*; *Jiang & Leung, 2005*). Successful expression hinges on accurate retrieval of representations. Transfer is typically linked to successful retrieval of old representations in new contexts, where similarity between initial learning and transfer events aids transfer (*Royer, 1979*). Retrieval studies highlight the importance of overlap between test and learning content for accurate retrieval (*Morris, Bransford & Franks, 1977*; *Roediger, Weldon & Challis, 1989*; *Tulving & Thomson, 1973*). Greater overlap increases the likelihood of activating correct representations. However, visual statistical learning

demonstrates extensive transfer across dimensions (*Turk-Browne & Scholl, 2009*), and contextual cueing in the visual domain can transfer to haptic search, suggesting a lack of specification of retrieval cue dimensions during statistical learning.

In line with us, *Turk-Browne, Jungé & Scholl (2005)* emphasized the critical role of selective attention in visual statistical learning. However, their argument implied that selective attention determines the information used for establishing abstract representations to mitigate the impact of changes in surface features. This reasoning is flawed because determining which information aligns with abstract rules presupposes the completion of abstracting core principles. Given our conclusion that representations are stimulus-specific and expression is flexible, we contend that selective attention should influence the phase of expressing learned representations rather than learning. Although the cognitive system cannot choose what to learn, it can decide which information to retrieve accordingly.

Similar to the design of our Experiment 2, *Jiang & Song (2005)* also required participants to learn two sets of repeated scenes with distinctive features in the learning phase. However, they failed to observe transfer in the testing phase, indicating a lack of flexibility in representation expression (Experiments 2 and 4). This outcome contradicted the conclusions drawn from our Experiment 2, which we attribute to their small sample size. However, further research is warranted to explore other possibilities. In their study, the testing phase unified the distinctive features into one. This operation maintained consistency in identity and spatial layout between the learning and testing phases for half of the repeated scenes in their study, allowing old retrieval cues to still facilitate performance. However, it also misled the cognitive system into expecting another set of scenes that did not appear.

Another finding of *Jiang & Song (2005)* was that participants exhibited contextual cue effects even when the identity changed (Experiments 1 and 3), consistent with *Chun & Jiang (1998)*. These findings underscore the role of spatial position in contextual cue effects. While our study does not refute this, the confirmation of expression flexibility raises questions about whether spatial position is an essential factor in the expression, warranting further investigation.

It is noteworthy that contextual cue representations encompass not only stimulus-related information but also attentional involvement, as attentional guidance is a crucial component of contextual cue expression (*Chun & Jiang, 1998*; *Jiang & Chun, 2001*; *Sisk, Remington & Jiang, 2019*). While our study focuses on visual representations and their associated activities, it does not deny that attentional guidance processes may also contribute to contextual cue representations. *Zang et al. (2022)* manipulated attention allocation during the learning phase by setting half of the items in the same repeated scene as white and the other half as black, cueing the target color at the beginning of the trial. They found that after losing the pre-cueing cues in the testing phase, the contextual cue effect disappeared. Pre-cues can initiate attention allocation processes in advance (*Fazekas & Nanay, 2017*; *Hede, 1980*), thus this result can be interpreted as the

failure of representation retrieval in attention processes due to the loss of pre-cues. This study suggests that in addition to visual representations, contextual cues also involve attention-related representations. However, further research is needed to explore this issue.

## CONCLUSIONS

The objective of this study was to investigate the characteristics of contextual cue representations and their expression. We posited that contextual cue representations are stimulus-specific, and their expression process is flexible. Our findings demonstrated that maintaining the spatial positions of distractors within repeated scenes while randomly altering their non-spatial identity impeded contextual cue learning. This underscores the significance of identity in contextual cue representations, confirming their stimulus-specific nature. Additionally, regardless of whether the expression of contextual cue representations was interfered with during the learning phase, an equivalent degree of transfer was observed after swapping the distinctive feature. This suggests that the cognitive system flexibly selects retrieval cues, highlighting the flexibility in representation expression.

This study has raised several unresolved issues worthy of further investigation. Firstly, it pertains to the extent to which expression can adapt to changes in environmental stimuli following the introduction of expectations regarding task regularities. Secondly, it prompts a reevaluation of the significance of spatial location in contextual cue expression. Lastly, it calls for exploration into how attentional mechanisms are involved in contextual cue representations.

## ACKNOWLEDGEMENTS

Thanks to my mother, Ms. Wenqin Zhang, for her selfless love.

### Funding

This work was supported by the Guizhou Provincial Science and Technology Foundation Program (Grant No. Qiankehe Jichu-ZK[2023]General 275). The funders had no role in study design, data collection and analysis, decision to publish, or preparation of the manuscript.

### Grant Disclosures

The following grant information was disclosed by the authors:
Guizhou Provincial Science and Technology Foundation Program: Qiankehe Jichu-ZK [2023]General 275.

### Competing Interests

The authors declare that they have no competing interests.

## Author Contributions

- Xiaoyu Chen conceived and designed the experiments, analyzed the data, prepared figures and/or tables, authored or reviewed drafts of the article, and approved the final draft.
- Shuliang Bai performed the experiments, authored or reviewed drafts of the article, and approved the final draft.
- Qidan Ren performed the experiments, authored or reviewed drafts of the article, and approved the final draft.
- Yi Chen performed the experiments, authored or reviewed drafts of the article, and approved the final draft.
- Fangfang Long conceived and designed the experiments, authored or reviewed drafts of the article, and approved the final draft.
- Ying Jiang conceived and designed the experiments, authored or reviewed drafts of the article, and approved the final draft.

## Human Ethics

The following information was supplied relating to ethical approvals (*i.e.*, approving body and any reference numbers):

The School of Psychology, Guizhou Normal University granted ethical approval to carry out the study within its facilities.

## Data Availability

The data is available at OSF: Chen, Xiaoyu. 2023. "Item Identity in Contextual Cueing." OSF. November 30. osf.io/6qfne.

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
