# Peer review of "The representation of contextual cue is stimulus-specific yet its expression is flexible"

_PeerJ, doi:10.7717/peerj.17318_

## Round 0.1 · original submission · Major Revisions

Thank you for submitting your manuscript to PeerJ. I have now received reviews from two experts (Tom Beesley has submitted a signed review as Reviewer 2). I have also read over the manuscript myself. You will see that the reviewers both acknowledge that your work makes an interesting contribution to the contextual cuing literature, and I agree. However, they both raise concerns about the integration of the current work within the broader literature that need to be addressed.

I am returning your manuscript, but invite you to submit a revision that addresses the issues raised by the reviewers. I detail below what I perceive to be the most salient points to address in this revision. I am hoping to avoid a protracted review process, and will aim to make a final decision on the next iteration of the manuscript. In preparing your revision, please ensure that you address all of the reviewer comments. I intend to send the revision out to the same reviewers for comment.

Both reviewers comment on having difficulty interpreting your expression at times. Reviewer 2 provides some specific examples where terminology could be changed to adopt more conventional phrasing (e.g., referring to encoding and retrieval processes). I had similar thoughts with respect to the “interference expression” and “non-interference expression” nomenclature, thinking that these might be described more simply using standard “No Load” and “Load” terminology. As a final comment on presentation, I found that there was a fair amount of repetition of some of the arguments and findings that could be streamlined to minimize redundancy.

Reviewer 1 notes several other findings that conflict with the results of your Experiment 1, which requires some discussion. They also suggest an interesting two-phase version of the experiment that might help clarify factors that differentiate establishing the representations underpinning contextual cuing from the expression of the effect later in the task. I do not think the additional experiment is necessary—though it would likely prove valuable—but the design should minimally be discussed as a logical next step for future research.

Both reviewers point out the similarity between your Experiment 2 and Experiment 2 of Chun and Jiang (1998). While there are certainly merits from conducting close replications, you do not introduce Experiment 2 as such. Given the absence of a clear rationale for Experiment 2 (over and above what can be gleaned from similar previous work), there is a question over what can be learned from Experiment 2. The motivation for Experiment 2 needs to be clarified, particularly with respect to what novel information can be taken from it.

Last, Reviewer 2 notes that there are instances where you argue from a null result. Given the relatively small sample sizes, there are concerns that non-significant results could be due to insufficient statistical power. Some discussion of this issue is needed. Reviewer 2 also advocates for the use of Bayesian statistical analyses, which would be appropriate, especially in cases of arguing in favor of a null result.

I thank you for submitting your manuscript to PeerJ and thank the reviewers for their insightful comments on your work. I look forward to receiving a revised version of this manuscript.

Yours sincerely,

David K Sewell

Reviewer 1 ·

Basic reporting

The language is difficult to follow at times.

Experimental design

Please see general comments below.

Validity of the findings

Please see general comments below.

Additional comments

In their study, the authors investigated contextual cue representations, focusing on whether these cues include distractor identity information and their flexibility in expression.

Experiment 1 examined if contextual cue representations are stimulus-specific by altering distractor identity consistency in repeated scenes. The results showed a significant reaction time difference only under the identity consistent condition, not when the identity changed. The authors interpret this as evidence that distractor identity is crucial to contextual cue representations, confirming their stimulus-specific nature.

Experiment 2 tested the flexibility of these representations. Participants were exposed to repeated scenes with variable colors under different conditions. Despite color interchange in the testing phase, the unchanged contextual cueing effect led the authors to conclude that the cognitive system can flexibly adapt the information used as retrieval cues. This supports their hypothesis of high flexibility in the expression of contextual cue representations.

The manuscript under review presents an interesting investigation into stimulus identity's role in forming context-target associations. However, several concerns regarding its alignment with existing literature, novelty of findings, and the experimental design need to be addressed.

In Experiment 1, the authors omit a crucial reference to Chun and Jiang's seminal work from 1998, particularly regarding Experiment 2. This omission is notable because Chun and Jiang demonstrated that the identity of items does not significantly impact the expression of learned memory traces once context-target associations are established. This finding contrasts with the results of the current study, where varying the identity of stimuli (orientations) prevented the formation of context associations, suggesting a dependency of context-target associations on the identity of distractors. A more robust approach for the authors could have been implementing a two-phase design in Experiment 1, with a learning phase featuring either fixed or variable configurations of distractors and an opposite configuration in the testing phase. This would have provided clearer insights into how distractor identity influences the establishment versus the expression of the cueing effect.

Additionally, the current findings align with the probability cueing concept explored by Zinchenko et al. (2018), where a higher percentage of environmental noise leads to no cueing effect. This raises the possibility that the current study's findings might be more about probability and noise rather than specific identity learning, i.e., variability in repeated displays adds to the uncertainty about the environment, thus limiting the cueing effect.

Finally, for Exp. 1, I think the work of Zang et al. (2022), AP&P is relevant. Zang et al. (2022) also found that task-irrelevant identity information is not learned and results in no cueing effect unless made task-relevant, which could further inform the current interpretation.

Regarding Experiment 2 in the manuscript, the authors make specific details and assumptions that merit closer examination and critique. The authors designed Experiment 2 to investigate the flexibility in contextual cueing (CC) expression. They introduced a visuospatial working memory (WM) task alongside the primary visual search task, assuming that this additional WM task would hinder the CC effect. This assumption, however, is challenged by recent findings in the field. Specifically, the study by Vicente-Conesa et al. (2022) provides compelling evidence from a large sample that an additional visuospatial task does not, in fact, hinder the contextual cueing effect. This raises questions about the efficacy and relevance of the triple task manipulation employed in Experiment 2 of the current study. If the addition of a visuospatial WM task does not impact CC as previously thought, then the manipulation in Experiment 2 might not provide the intended insights into the flexibility of CC. Instead, it might only serve to increase the overall difficulty of the task without offering meaningful data on how context learning is affected.

Furthermore, the design of Experiment 2 includes a color change in the testing phase, presumably to investigate how changes in visual features impact the retrieval of contextual cues. However, this approach seems to replicate the logic of Chun and Jiang’s Experiment 2 from 1998, where a similar manipulation was used. In their study, Chun and Jiang found that changes in the color of distractors did not significantly impact the expression of learned contextual cues, suggesting a certain degree of flexibility in the CC effect concerning changes in visual features. Given that this effect is already well-documented, the color change manipulation in the current study's Experiment 2 does not appear to provide new insights into the flexibility of CC. Instead, it seems to reaffirm a well-established understanding of the phenomenon. In light of these observations, Experiment 2 of the current study might benefit from a redesign that more directly and innovatively probes the flexibility of contextual cueing, possibly by incorporating different types of manipulations that have not been extensively explored in previous research. This could help in distinguishing the study's contributions from the established findings and offer fresh insights into the nature of contextual cueing.

·

Basic reporting

Some issues with the language used. Should be past tense throughout.

Raw data has been shared. Analysis scripts are not available - analysis procedure cannot be verified.

Experimental design

No issues.

Validity of the findings

Experiment 1 is clearly of use. There is a question of novelty over Experiment 2 - that is, if this is a replication, then that has not been sufficiently justified.

Additional comments

Two experiments examined the role that distractor identity plays in the encoding and retrieval of representations in contextual cuing of visual search. In Experiment 1, the repeated configurations were either trained with consistent orientations, or inconsistent orientations (rotated on each trial). It was found that CC only emerged for consistent orientations, suggesting that CC requires fixed distractor identities for encoding of the predictive information in the display. In Experiment 2, participants were trained with consistent distractor features in one colour, and then tested with a transfer to a different feature set (red to green; green to red). CC was observed in this transfer test. Participants completed this task under load and no-load conditions; this manipulation did not make a different to the CC observed. Experiment 2 is purported to show intact expression of CC and therefore the flexibility of the performance from the acquired representation.
The topic is of interest to people in the fields of visual search, associative/statistical learning, etc. I have some concerns about the usefulness of Experiment 2, given the similarity to previous experiments. I think a clearer case might be made for how these data should be interpreted within the existing literature.

Chronological points:
1. The last paragraph of the intro is not clear to me. Particularly “the formation of a fixed expression mode” (L 148) – it’s not clear to me what this means in terms of cognitive processes. I have to admit I struggled at times to gain a clear picture of what the authors were saying about (what are to me at least) the processes of encoding and retrieval. At times I felt the somewhat verbose expression could lead to confusion (lines 131-135 are a good example).
2. The results of Experiment 1 are clear. I’m not aware of a paper that has done this manipulation before (inconsistent identities during training), which is a little surprising – it’s such an obvious design. The authors note the Markovski (2016) paper, and the rationale for the current study is reasonable: those experiments used slightly unusual stimuli with semantic content and it is useful to ask if the same effect is observed in a more standard CC task. They might also reference Markovski (2017), which seems relevant.
3. Lines 287-293: The authors wish to show rapid CC across the initial period of training. Rather than rely on a null result, it might be best to analyse at the sub-epoch level in the first epoch of two. That is, divide the epochs into 2 or 3 “blocks” and show the emergence of CC (the interaction with the time factor).
4. I have more issues with the usefulness of Experiment 2. Firstly, it’s not clear what the logic is for employing a manipulation of working memory load. The author’s state: “If both interference and non-interference expression conditions exhibit similar magnitudes of contextual cue effects during the testing phase, it suggests that prior experiences do not affect the subsequent cognitive system's flexible selection of retrieval cues. The cognitive system can selectively choose partial information within the scene as retrieval cues based on the immediate requirement to optimize representation retrieval efficiency.” I have to admit I’m a bit lost here with the logic – why is it necessary to show equivalent CC under WM load in order to draw this conclusion? Are you making a statement about the automaticity of the processes?
5. It seems to me that a simpler test of the flexibility of the retrieval would have been to compare tests in which the surface features change, to conditions where they don’t change. But of course this has already been done quite a bit. The authors cite a few instances, but note also Endo and Takeda (2004), and I believe the original Chun and Jiang (1998) paper showed this in their Experiment 2. There is therefore a question mark over the novelty and usefulness of the current Experiment 2.
6. In my opinion, Experiment 2 is limited in what it can tell us, since there is no condition in which the surface level features do not change. Therefore in this design we don’t know to what extent the surface features are important for intact CC. This is problematic for some of the author’s reasoning in the discussion: “whether the expression of the contextual cue representation during the learning phase was interfered with or not, it did not affect the degree of transfer of contextual cue representation in the testing phase” [my italics] – critically here, we cannot know the degree to which it affected the CC observed, since we do not have the relevant comparison (to a no transfer condition).
7. Line 493-495. The authors would like to draw conclusions from the non-significant result. This would be better supported with a Bayesian analysis to test the evidence in support of the null result.
8. Lines 592- 596: “Given that Experiment 1 confirmed the presence of identity information in the contextual cue representations, the transfer observed in Experiment 2 cannot be attributed to the failure to learn color information in the repeated scenes. Instead, it suggests that in the testing phase, the cognitive system did not consider scene color as part of cue for retrieval.” I’m not sure I follow the logic here. Colour and object orientation are two different features of the identity of the stimulus. Experiment 1 shows that orientation is important for encoding of the configuration. But why does that then mean that colour must have been encoded? It could be that orientation, but not colour, is an important feature for encoding. We do not have the correct experimental conditions to determine if this is true.

Signed Tom Beesley

---

## Round 0.2 · accepted · Accept

I invited both of the original reviewers back. Reviewer 1 is happy to accept the revised manuscript. I did not hear back from Reviewer 2, however, but having looked over the revision and considering their original comments, I am confident that these have been successfully addressed.

One thing that could be added to clarify the Bayesian t-test results on Lines 431-432 is to note what classification scheme you are relying on to interpret the Bayes Factors (e.g., Lee and Wagenmakers, 2013, report an adaptation of the classification scheme used by Jeffreys, 1961).

I also urge you to perform one last thorough proofreading of the manuscript. For example, Line 60 is missing a word, "the contextual [cueing] effect". The bulleting of points in the Results section of Experiment 2 could also be expanded. Currently, you use numbered bullets, but this deviates in this part of the manuscript.

Once these quite minor changes are made, the manuscript will be ready for publication. Congratulations!

Reviewer 1 ·

Basic reporting

Acceptable

Experimental design

Acceptable

Validity of the findings

Acceptable

Additional comments

I appreciate the authors' revision of the manuscript and I do not have any further comments or questions.